# Quasi-3D Mechanistic Model for Predicting Eye Drop Distribution in the Human Tear Film

**DOI:** 10.3390/bioengineering12080825

**Published:** 2025-07-30

**Authors:** Harsha T. Garimella, Carly Norris, Carrie German, Andrzej Przekwas, Ross Walenga, Andrew Babiskin, Ming-Liang Tan

**Affiliations:** 1Biomedical, Energy, and Materials Division, CFD Research Corporation, 6820 Moquin Drive NW, Huntsville, AL 35806, USA; carly.norris@cfd-research.com (C.N.); carrie.german@cfd-research.com (C.G.); andrzej.przekwas@cfd-research.com (A.P.); 2Division of Quantitative Methods and Modeling, Office of Research and Standards, Office of Generic Drugs, Center for Drug Evaluation and Research, U.S. Food and Drug Administration, 10903 New Hampshire Avenue, Silver Spring, MD 20993, USA; ross.walenga@fda.hhs.gov (R.W.); andrew.babiskin@fda.hhs.gov (A.B.); mingliang.tan@fda.hhs.gov (M.-L.T.)

**Keywords:** tear film, dexamethasone, quasi-3D, eye drop, fast-running, blinking, drainage

## Abstract

Topical drug administration is a common method of delivering medications to the eye to treat various ocular conditions, including glaucoma, dry eye, and inflammation. Drug efficacy following topical administration, including the drug’s distribution within the eye, absorption and elimination rates, and physiological responses can be predicted using physiologically based pharmacokinetic (PBPK) modeling. High-resolution computational models of the eye are desirable to improve simulations of drug delivery; however, these approaches can have long run times. In this study, a fast-running computational quasi-3D (Q3D) model of the human tear film was developed to account for absorption, blinking, drainage, and evaporation. Visualization of blinking mechanics and flow distributions throughout the tear film were enabled using this Q3D approach. Average drug absorption throughout the tear film subregions was quantified using a high-resolution compartment model based on a system of ordinary differential equations (ODEs). Simulations were validated by comparing them with experimental data from topical administration of 0.1% dexamethasone suspension in the tear film (R^2^ = 0.76, RMSE = 8.7, AARD = 28.8%). Overall, the Q3D tear film model accounts for critical mechanistic factors (e.g., blinking and drainage) not previously included in fast-running models. Further, this work demonstrated methods toward improved computational efficiency, where central processing unit (CPU) time was decreased while maintaining accuracy. Building upon this work, this Q3D approach applied to the tear film will allow for more seamless integration into full-body models, which will be an extremely valuable tool in the development of treatments for ocular conditions.

## 1. Introduction

Topical ocular administration refers to the application of a drug to the surface of the eye, typically in the form of eye drops or ointments. Topical administration is a common route of drug delivery in the eye and is used to treat a wide range of ocular conditions, including glaucoma, dry eye, and inflammation, which can lead to chronic outcomes and visual disturbances if not effectively managed [1,2]. There are numerous physical barriers that limit the bioavailability of topical drugs applied to the ocular surface, restricting the efficacy of many potential treatments for these conditions [3]. Efforts to improve ophthalmic drug delivery and to defeat these barriers have led to an increase in innovative solutions for targeted treatments in the field of ophthalmology using nanotechnology-based formulations [3,4]. Nevertheless, additional complications affecting pathological ocular conditions, particularly for dry eye, remain a research challenge. Such complications include poor maintenance of the tear film due to blinking and tear drainage post-administration [5,6]. Further, the U.S. Food and Drug Administration (FDA) mandates that generic drugs are shown to be bioequivalent to branded drugs [7], yet tests for topical ophthalmic drug bioequivalence are challenging due to the complexity of the ocular system and a lack of appropriate and sensitive testing methods to demonstrate the bioequivalence between branded and generic drug products in humans. To help address these challenges, computational strategies accounting for complex tear film anatomy, blinking, and drainage may provide a mechanistic approach to understanding drug absorption and exposure in ocular tissues that may be used to study the bioavailability and bioequivalence of topically administered ophthalmic drug products.

Computational modeling can be a useful tool for studying the behavior of drugs after topical administration in the eye, for testing different treatment regimens, and for assessing the potential for adverse reactions [8,9,10,11,12,13]. Physiologically based pharmacokinetic (PBPK) modeling, specifically, ordinary differential equation (ODE)-based compartment models, are most commonly used to predict the absorption, distribution, metabolism, and elimination of drugs, streamlining research approaches toward solutions with high efficacy [14]. The ocular anatomy as represented in PBPK models is typically divided into several anatomical compartments, and a set of ODEs is used to describe the movement of drugs between compartments [15,16]. However, these models neglect structural geometry, thereby simplifying the physiological and geometrical characteristics of the eye. Using a compartmental ODE modeling approach, the spatiotemporal segment-specific dynamics of ocular drug absorption cannot be captured. Specifically, the dynamic effects of blinking on tear drainage or drug redistribution on drug bioavailability in the tear film cannot be accounted for. Thus, the implementation of higher-resolution models accounting for spatiotemporal distribution is anticipated to more accurately predict eye drop distribution following topical administration.

A number of spatial computational models have been previously reported, such as partial differential equation (PDE)-based and finite element model (FEM) ocular models, with the ability to account for ocular fluid flow and drug distribution in anatomically resolved subregions of the eye [8,10,11]. However, this added complexity typically necessitates significant computational resources and expertise, which can be a limiting factor in their widespread adoption and application in clinical settings. To overcome these modeling barriers, quasi-3D (Q3D) modeling can be employed as an efficient, fast-running, PDE-based alternative used to solve both hydrodynamics and species (i.e., particles, dissolved drug, excipients) transport equations. The major advantages of this approach are the ease of model setup, high computational speed, simple visualization of results, and ease of linking to compact (i.e., spring/mass/damper devices, valves, pumps, etc.) and compartmental models. A detailed table comparing PBPK, FEM, and Q3D modeling strategies can be found in the Appendix A.

Combined Q3D–compartment modeling approaches were previously implemented for alternative applications. Kannan et al. initially implemented a combined approach for the purpose of modeling regional lung constriction in healthy and diseased lungs to optimize the effects of orally inhaled drug products [17,18,19,20]. The spatially resolved absorption kinetics of drug inhalation were simulated using a Q3D mesh capturing the tubular multilayer structure of the trachea, non-symmetrical bifurcations and out-of-plane rotations of the bifurcations, and non-circular cross-sections, among other critical features of the pulmonary system. Beyond the region of interest, anatomy was treated as one-dimensional compartments linked to whole-body PBPK simulations, achieving accurate results up to 25,000 times faster than alternative 3D methods. Using a similar approach, a Q3D model of the cornea was developed to simulate drug transport across the in vitro cornea, where the model was shown to provide both efficient and accurate predictions of lipophilic (RhB) and hydrophilic (fluorescein) tracer distribution throughout the cornea [21]. However, the accuracy of predictions of corneal distribution are directly dependent on the accuracy of tear film distribution. This can be especially important when investigating the effects of ocular conditions on drug distribution. Building upon this foundational work, the purpose of this study was to develop a fast-running, multiphysics Q3D-based model of the tear film to simulate drug administration, drainage, tear evaporation, and spatial drug distribution, accounting for blinking effects. This Q3D-based computational model can be linked to full-body integrated PBPK models predicting drug behavior, intended to accelerate drug discovery and clinical testing, improve efficacy, and reduce the likelihood of adverse reactions among patient populations.

## 2. Materials and Methods

### 2.1. Approach

The unique physio-anatomy of the eye constitutes static and dynamic factors affecting ocular drug delivery [22]. Upon administration of the eye drop to the ocular surface, the blinking reflex is triggered by the increase in tear volume, eliminating most of the volume applied [23]. The remaining drug solution mixes with the tear film, which is subject to clearance via nasolacrimal drainage or evaporation [24]. For these reasons, only a fraction of the drug permeates the corneal surface and surrounding ocular structures. To accurately capture the factors influencing drug distribution from topical administration, a Q3D model was developed to account for blinking, drainage, and evaporation of the tear film. The Q3D tear film model was then integrated with an anatomically resolved compartmental model of the whole human eye to simulate species transport. All simulations were conducted using Computational Biology (CoBi) Tools, a commonly used framework for applications in bioengineering and pharmacology [20,21,25,26,27,28,29,30,31]. Standard solvers, such as Euler method, Runge–Kutta, and Stiff solvers are implemented into CoBi software (developed by CFD Research; accessed May 2025) to solve systems of ODEs. The platform provides multiple solver options including explicit methods (Euler, Runge–Kutta) for non-stiff systems and specialized stiff solvers for systems with widely varying time scales. Optimization of Q3D tear film dynamics was conducted to ensure agreement with previously published models [21]. Lastly, validation of species transport in the tear film was performed through simulation of the dexamethasone (Maxidex^®^) concentration–time profile using the combined Q3D–compartment model.

### 2.2. Tear Film Anatomy

The tear film is a thin layer of moisture that covers the surface of the eye (Figure 1A). Although it is often thought of as a single entity, the tear film is composed of three distinct regions: the upper and lower fornical sacs, the upper and lower menisci, and the tear film. The fornical sacs (also known as conjunctival sacs) are small pockets of tears in the fornices, or folds, of the eyelids (Figure 1B; yellow). The bulbar conjunctiva is the region adjacent to the anterior sclera, and the palpebral conjunctiva is the region lining the eyelid (Figure 1B; green). Fluid is collected in the fornical sac, and excess tears are then drained away through canaliculi channels to the lacrimal sac near the inner corner of the eyelid. The menisci (Figure 1B; red) are layers of tears that form at the inner corners of the eyelids, acting as a reservoir that is distributed evenly over the tear film during blink, preventing it from draining away too quickly. The tear film region (Figure 1B; blue) is the thin layer of moisture that covers the exposed surface of the eye. It is composed of lipid, aqueous, and mucin layers. High-resolution species transport within the tear film was implemented according to these defined anatomical regions, and the flow assumptions are depicted in Figure 1C.

### 2.3. Q3D Design and Assumptions

Construction of the Q3D geometry and structural mesh was based on methods previously described by Pak et al. [21] simulating drug transport through the in vitro cornea. Similarly, to approximate the tear film geometry, 3D rectangular prisms were discretized and combined to form a ducting network (Q3D segments) representing five regions of the tear film: upper and lower fornical sacs, upper and lower menisci, and the pre-corneal tear film. Mesh sizes (10–20 Q3D segments) were optimized to enhance visual resolution and capture concentration gradients. A separate cylindrical network was formulated to represent the canaliculi, where drainage occurs from the menisci. Geometric assumptions are provided in Table 1 and depicted in Figure 2.

Q3D tear film design, assumptions, and flow distributions are shown in Figure 2. The convective–diffusive transport processes in a Q3D region are described by PDEs (Appendix A), which were solved numerically in the mesh-resolved direction and analytically in the transverse direction. Discretization of the tear-film network geometry enabled the capture of localized concentration gradients. The computational meshes in the upper and lower fornical sac and meniscus regions were oriented perpendicular to the tear film structural networks to more accurately represent the directional flow distribution to the cornea.

### 2.4. Modeling Blink Cycles

Tear film volume changes due to blinking were modeled based on three distinct phases of the blink cycle: interblink, closure, and deposition. During one full blink cycle, the eyes remain open and there is no movement of the eyelids (interblink phase; 5.5 s), the upper eyelid falls (closure; 0.04 s), and then the upper eyelid rises (deposition; 0.18 s) [37,40,41,42,43]. Together, closure and deposition make up the blink phase. Water and salt balances during the blink cycle were defined based on previously published equations and assumptions [34,44]. Verification of the dynamic changes in volume and osmolarity over the course of multiple blink cycles was performed in CoBi, and outputs were compared with previous models by Walenga et al. [34].

Computational performance of the Q3D tear film model was quantified by the central processing unit (CPU) time and optimized by periodically cutting the integration time during the blinking period and allowing longer timesteps during the interblink period. An initial timestep for the blink period of dt = 0.001 s was selected by the user. CPU times were compared to determine ideal timestep parameters and assess computational performance compared with alternative methods.

### 2.5. Modeling Tear Fluid Drainage

Tear fluid is produced by the lacrimal glands and is eliminated mainly by drainage through the canaliculi, which contributes to nearly 50% of tear fluid elimination [45]. Drainage is an active process closely tied to the blink cycle. The cyclic downward and upward motion of the eyelids results in a pressure gradient across the canaliculi connecting the meniscus and lacrimal sac. During the blink phase, the lid closure acts to squeeze the canaliculi to drain the fluid into the lacrimal sac. During the interblink phase, when the eyelids are in an open state, the vacuum created during the previous step results in tear fluid flow from the meniscus into the canaliculi. The rate of change in tear volume caused by drainage ($Q_{drainage}$) can be determined as a function of the difference in tear volumes over the blink cycle (Equation (1)) where $V_{blink}$ and $V_{interblink}$ are the canaliculi volumes at the end of the blink and interblink phases, respectively.(1)$$\mathrm{Drainage}\;\mathrm{Rate}:\;Q_{drainage}=\frac{V_{\mathrm{int}erblink}-V_{blink}}{t_{blinkcycle}}$$

The volume of tear fluid that drains into the canaliculi is a function of the rate of change in canaliculi radius (Equation (2)), where *R* is the radius of the canaliculi, *bE* is the product of canalicular wall thickness and elastic modulus, *R*_0_ is the undeformed radius of the canaliculi, and *μ* is the viscosity of the instilled fluid. Canaliculi radius is particularly sensitive to the parameter assumption of *bE*, which has been estimated to range from 0.649 Pa-m to 34.022 Pa-m [46]. Because the canaliculus is a viscoelastic material with frequency-dependent moduli, this parameter is difficult to estimate in humans. In this work, *bE* was assumed to be 2.57 Pa-m based on the elastic zero frequency mechanical properties of the porcine canaliculus [47].

Upon topical administration of eye drops, the viscosity of the tear fluid mixture may change from Newtonian to non-Newtonian. In the case of non-Newtonian fluids, *n* and *K* are rheological parameters that can be obtained by fitting the viscosity flow curve (viscosity vs. shear rate). For a Newtonian fluid, *n* is equal to 1. Drainage parameter assumptions governing the change in radius of the canaliculi are provided in Table 2.(2)$$\mathrm{Change}\;\mathrm{in}\;\mathrm{Canaliculi}\;\mathrm{Radius}:\;\frac{\partial R}{\partial t}=\frac{1}{2}{\left(\frac{bE}{2K{R_{0}}^{2}}\right)}^{\frac{1}{n}}\frac{{R_{0}}^{\frac{2n+1}{n}}}{3n+1}{\left(-\frac{\partial R}{\partial x}\right)}^{\left(\frac{1}{n}-1\right)}\frac{\partial^{2}R}{\partial x^{2}}$$

The change in canaliculi radius over the blink cycle and the associated drainage rate of tear fluid into the canaliculi were implemented into CoBi software (developed by CFD Research; accessed May 2025) using a finite difference approximation. Steady-state assumptions are provided in Table 3. Estimation of drainage rates and canaliculi radii were then verified against previously published physics-based drainage models [46,48].

### 2.6. Modeling Tear Fluid Evaporation

The evaporation flow rate was calculated as a time-varying profile using a tear film breakup time model previously described by Walenga et al. [34]. The pressure relationship to the evaporative mass flux was calibrated to 0.1 K/Pa, and the pressure and temperature relationship to the mass flux was estimated as 3 × 10^4^ K-m^2^-s/kg, corresponding to an evaporation rate of approximately 0.2 µL/min. This evaporation rate was selected based on the literature citing mean evaporative rates of 0.2 µL/min in patients with dry eye [49].

### 2.7. Tear Film Species Transport Validation

The Q3D human tear film was linked to a whole-eye compartmental model describing species transport to enable assessment of model performance. The compartmental model of the whole human eye was adapted from German et al. [25,27] and developed according to species transport assumptions depicted in Figure 3. All regions were included as homogenous compartments apart from the tear film, menisci, and fornical sacs (Figure 3; blue). A high-resolution model of the cornea was included, accounting for epithelium, stroma, and endothelium layers. Additional compartments included the conjunctiva, lens, iris-ciliary body, aqueous humor, vitreous body, choroid, sclera, and retinal layers. Systems of equations describing species transport are provided in Appendix A.

Species transport of a 0.1% dexamethasone suspension (Maxidex^®^) applied topically in the tear film was simulated and validated using previously published datasets. Tear film volume changes due to eye drop instillation were accounted for in the tear film and menisci compartments according to their basal volume ratios. This initial condition can be calibrated to reflect alternate conditions, such as full application of the eye drop to the lower fornical sac. Volume in excess of the maximal volume capacity for the human eye (25 µL) was assumed to be lost to spillover [34], including drug mass associated with the excess volume.

The Maxidex^®^ suspension contains cosolvents to enhance drug solubility. While the solubility of dexamethasone in water at 25 °C is approximately 0.089 mg/mL [50], this solubility was assumed to be closer to 0.1 mg/mL in the presence of excipients [51]. Average particle radius was assumed to be 1.5 µm, and distribution coefficient was estimated to be 2.5, based on previously reported estimations [52,53,54,55]. Drug-specific inputs are provided in Table 4, and calibrated parameters are reported in Appendix A.

Validation data were published in Jóhannesson et al., where 0.1% dexamethasone suspension (Maxidex^®^) concentration was quantified in the tear film of six healthy volunteers at 0.25, 0.5, 1, 2, 4, 6, and 24 h following topical administration. A micropipette capillary tube was placed in the tear meniscus inside the lower eyelid to collect approximately 5 µL of tear fluid [56]. Initial concentration was assumed to be zero, and validation was performed within the first six hours following administration based on reports that dexamethasone concentrations were approximately zero by 24 h. Model fit between the concentration–time profile simulated in the tear film and published experimental concentrations was determined using the coefficient of determination (R^2^) method, and simulation error was quantified using RMSE and AARD measures.

## 3. Results

### 3.1. Verification of Blink Cycle Volume and Osmolarity Dynamics

Blink cycles were implemented into the Q3D tear film model. Over the course of one blink cycle (5.72 s), changes in the upper meniscus, tear film, and lower meniscus volumes followed anticipated visual hydrodynamics (Figure 4). At the beginning of the interblink phase, tear fluid produced by the lacrimal glands enters the upper and lower conjunctival sacs, flows to the lid margin, and enters the upper and lower meniscus compartments. By the end of the interblink phase, tear fluid within the tear film undergoes punctal duct drainage and evaporation, resulting in a decreased tear film volume. During the closure phase, the upper eyelid descends toward the lower eyelid, causing the tear film to mix completely between all compartments. Lastly, the upper eyelid rises during the deposition stage, leading to the homogenous redistribution of the tear film, which results in shrinkage of the meniscus.

Following verification of the blink cycle hydrodynamics, changes in volume (Figure 5A) and osmolarity (Figure 5B) for the meniscus and tear film regions over the course of 20 blink cycles were simulated in CoBi. Results were verified against a compartmental model previously developed by Walenga et al. [34]. Q3D volumes and osmolarities matched previous datasets, verifying effective implementation.

### 3.2. Assessment of Computational Performance

Computational performance using variable and periodic time stepping was performed to optimize the computational efficiency of the Q3D tear film model. Seven test cases were conducted in CoBi, defined in Table 5. All simulations were performed on personal computing laptops (Dell) with 64-bit Windows operating systems. CPU was computed on devices equipped with 16 GB of RAM. Please note that the CoBi modeling platform is OS-agnostic and is not restricted to Windows operating systems or any specific hardware manufacturer. A uniform timestep value of 0.001 s was chosen at the beginning of the simulation for Case A, resulting in the greatest CPU time of 13 m 47 s. While this smaller timestep was necessary for the closure phase, the model could adopt a higher timestep value during the interblink phase (duration = 5.5 s), thus reducing the overall computational cost. Significant scale-down in the run time of the Q3D-based blinking human tear film model was observed from Case A to Case E with an optimal CPU time of 1 m 02 s. Cases F and G resulted in run times higher than that of Case E, despite the timestep during the interblink phase being higher. This is due to the total number of timesteps, as Case E required fewer timesteps per blink cycle (interblink and blink). Volume (Figure 6A) and osmolarity dynamics (Figure 6B) comparing timestep case results over 10 blink cycles demonstrated preservation of the model performance for each test case.

### 3.3. Verification of Nasolacrimal Drainage Rates

A physics-based approach to calculating nasolacrimal drainage approximated tear fluid volume loss as a function of the change in canaliculus radius during the interblink and blink phases, as defined by Zhu and Chauhan [46]. The model implemented steady-state assumptions obtained analytically and defined in Table 3, where the interblink phase approached a steady state value of Rb, and the blink phase radius reached a steady state value of Rib (Figure 7A). Based on these assumptions, nasolacrimal drainage rates were calculated from the difference in canalicular volume after the interblink and blink phases, ranging from 0.7 to 0.9 µL/min (Figure 7B).

### 3.4. Validation of Tear Film Species Transport

Pharmacokinetic profiles from all tear film subregions were averaged together to create a single tear film averaged profile, which was then compared with experimental drug concentrations in the human tear film. Drug dissolution was modeled up to 6 h following topical drug administration (Figure 8). Concentrations rapidly reached the solubility limit and then decayed slowly over the next few hours. Predicted concentrations were within the range of reported experimental values. Note that the variability in the experimental datasets is large, demonstrating limitations in experimental method sensitivity and/or patient-specific factors affecting dissolution.

## 4. Discussion

A quasi-3D, biofidelic model of the human tear film was developed and validated to simulate the transport of topically administered drugs. The model incorporates various state-of-the-art techniques and physics-based approaches to describe the complex mechanisms involved. The authors have introduced a novel Q3D approach for modeling and simulating the distribution of eye drops in the tear film, accounting for blinking, drainage, and evaporation. Demonstration of the model’s feasibility and performance, shown in this study, supports critical discussion of modeling limitations that should be addressed to accelerate discoveries in the field of ocular health.

*Hydrodynamics, volume, and osmolarity all demonstrated agreement with previous models, justifying the initial conditions used for species transport.* Specifically, hydrodynamic conditions were satisfied during each stage of blinking, and tear film behavior was verified by comparing volume and osmolarity dynamics over 20 blink cycles with previously validated models [34]. Direct comparison of performance demonstrated preservation of model implementation. Incorporation of this mechanistic model accounting for blinking and drainage in the tear film is novel because the Q3D approach allows for fast-running spatiotemporal estimations (Figure 4) while maintaining computational efficiency.

*Predicted tear film drainage was within physiological ranges.* The difference in canalicular volume after interblink and blink phases ranged from 0.7 to 0.9 µL/min (Figure 7). For a normal tear film, the tear drainage rate can vary from 0.1 to 4 µL/min, verifying the accurate implementation of tear fluid drainage [46]. Large variations in reported tear drainage rates may be attributed to subject age, size and shape of puncta, canaliculi thickness, or elastic modulus of the canaliculi [35,57]. Limitations in modeling canaliculi radius and underlying variability are likely to stem the assumption of *bE*. In this work, *bE* was assumed to be 2.57 Pa-m, approximated from porcine canaliculi mechanical properties [47]. In a later study by Zhu and Chauhan, a mathematical model was developed to compare the effects of *bE* on canaliculi radius and associated drainage [57]. It was found that a *bE* value of 0.64 Pa-m may be a more accurate approximation of human canaliculi viscoelastic properties. As such, sensitivity analysis of this Q3D blinking model to variable *bE* should be evaluated in future implementations. Nevertheless, this model demonstrates that the canaliculi radius reached a steady state within each blink phase, ranging from 0.240 to 0.248 mm, consistent with human cadaver measurements where the average puncta radius was 0.250 mm [58].

*Tear fluid evaporation was assumed to be constant.* The constant rate of evaporation was calibrated in this model to be 0.2 µL/min based on previous model assumptions and the literature data [34,49]. This assumption could lead to over- or under-estimation under dynamic physiological conditions. However, future iterations implementing physics-based models developed by Cerretani et al. could provide significant advantages. These physics-based models can account for multiple factors influencing tear fluid evaporation, such as molecular properties, system geometry, concentration profiles both in the liquid and in the surrounding vapor, and convective heat and mass transfer in the liquid and vapor phases [24,37]. Thus, future iterations of this model will incorporate physics-based evaporation estimation.

*Computational efficiency was achieved via periodic time stepping.* Periodic time stepping methods are typically employed in higher-order numerical models to reduce run times [59]. Computational efficiency with periodic time stepping was optimized to perform evaluations within approximately one minute, comparable to typical ODE solvers. The lowest CPU was achieved when timesteps were aligned with the stages of the blink cycle. Considering that finite element models of ocular mechanics can last for days, a one-minute run time is much preferred [60]. This further highlights the advantages of using Q3D mechanistic models.

*Simulated drug concentrations in the tear film fell within experimental ranges.* Dexamethasone formulations have been on the market for decades, so sufficient data in humans allowed for the effective validation and assessment of drug-specific model performance. Simulation of Maxidex^®^ suspension fell within the range of experimental tear film concentrations at each time point, where R^2^ was 0.76. This measure accounts for variability in experimental data; 0.6–0.9 is considered a good fit in PBPK models [61]. Variability between model predictions and average experimental concentrations may also be reduced in future iterations by accounting for tear film layer (mucin, aqueous, and lipid) properties during drug transport. Error measures of RMSE and AARD were 8.7 and 28.8%, respectively, which indicates that the model can predict within 8.7 µg/mL on average, which is less than the standard deviation of the experimental dataset. Note that despite these error values, this model for drug distribution throughout the tear film was evaluated only based on a dataset from six healthy adults, and more robust, population-based assessments of model performance are necessary prior to implementation of this tool in clinical settings.

*Q3D models have a few main limitations when compared with alternative modeling approaches.* Q3D models may be more complex and time-consuming to set up and run than ODE models, particularly for large or complex models requiring the solution of partial differential equations over a 3D domain. For analysis of tear film dynamics alone, FEBio, ANSYS, and COMSOL Multiphysics are known for their intuitive design features [62]. Q3D models require more detailed and specific data inputs to accurately represent 3D geometry and boundary conditions, which may be challenging if data are not readily available. The complex geometry of the human eye and minimal datasets quantifying its complex anatomical features, particularly in the tear film, led to an approach centered around conservation of volume within rectangular prism substructures. However, anatomical or physiological variability among patient populations (e.g., blink rate, tear volume) inherently limits the generalizability of this model beyond that of a healthy adult. Q3D tear film model performance is directly tied to the accuracy of the geometric and physiological parameter assumptions, which we expect will continue to converge as technological innovations improve the quality of experimental assessments over time.

*Assessment and validation of drug clearance throughout the posterior eye and whole-body systems could enable use of the developed model for prediction of systemic safety in future iterations.* Areas for future development include transport of tear fluid into the bloodstream, which can occur through the lacrimal glands, the blood–retinal barrier, or interaction between blood vessels and outer ocular regions [63]. Expansion of model complexity to include these transport mechanisms can enable optimization of topical formulations to reduce the likelihood of adverse events from ocular delivery. This capability is advantageous, especially for potent corticosteroids like dexamethasone, which can contribute to a wide range of systemic complications [64].

*One common limitation among models is the lack of available datasets for validation.* While this work demonstrates the novelty and feasibility of Q3D implementation within the tear film, further validation should be conducted to ensure reliable prediction of drug distribution throughout various regions of the eye. This will require a large amount of experimental or literature data, which is mostly available for animals rather than humans. Additionally, the developed Q3D model was built based on experimentally derived parameters from healthy adults. The lack of available datasets for children and elderly individuals as well as those with ocular surface disorders limits the applications of this model. To establish a well-defined verification and validation protocol and expand the clinical applications of these models, the authors are working on compiling all available data into a single open-source public repository. This repository, originally intended for https://eye.health-map.net/, is currently available upon reasonable request. With access to larger, open-source datasets, ocular modeling frameworks may be used to perform virtual bioequivalence studies to support the development of generic ophthalmic drug products.

*Overall, the use of this Q3D ocular model is recommended as an alternative to the ODE, 2D, and 3D model approaches due to several advantages demonstrated in this feasibility study simulating drug transport in the eye.* The Q3D approach was selected to balance computational efficiency with spatial accuracy for ocular drug delivery applications characterized by highly skewed aspect ratios, where one dimension significantly exceeds the others (e.g., thin tear film layers, corneal epithelium). Q3D models offer intermediate complexity between 2D and full 3D approaches, capturing essential 3D effects through dimensional reduction techniques while avoiding the computational burden of full 3D finite element discretization. While full 3D models provide the highest spatial fidelity, Q3D achieves comparable accuracy for geometries with skewed aspect ratios at significantly reduced computational cost, effectively capturing dominant transport phenomena (diffusion across tissue layers, convective flow in the tear film) without significant loss of physical accuracy. These features allow researchers to explore a wide range of scenarios and conditions without having to build a new model from scratch, such as adding a feature to model contact lens interactions to the existing high-resolution eye model. The versatility of this Q3D tool for studying a variety of phenomena is further demonstrated by its ease of integration with ODE models and its ability to simulate both continuous and discrete processes, such as drug transport and drug absorption at the cellular level. Implementation of the Q3D-compartment model of the blinking eye developed in this work can be extended to provide age-specific or personalized simulations based on imaging, anthropometry, or disease conditions in future studies to demonstrate its full capabilities.

## 5. Conclusions

A novel quasi-3D (Q3D) model incorporated computational fluid dynamics within the tear film of the eye where computational efficiency was achieved using periodic time stepping within the interblink phase. Outputs were then integrated with a PBPK high-resolution compartment model of the human eye, and spatial drug distribution at the point of application was demonstrated to predict absorption in an efficient manner, thus addressing the need for high-resolution models accounting for spatiotemporal distribution to more accurately predict eye drop distribution following topical administration. Q3D tear film modeling could be a powerful tool for clinicians to recommend optimal dosing based on patient-specific parameters (i.e., geometry, drainage, blink rates) as a step toward improved personalized medicine. This robust approach will enable further extension of the model framework toward acceleration of drug discovery and clinical testing, improved efficacy, and reduced likelihood of adverse reactions among patient populations.

## Figures and Tables

**Figure 1 bioengineering-12-00825-f001:**
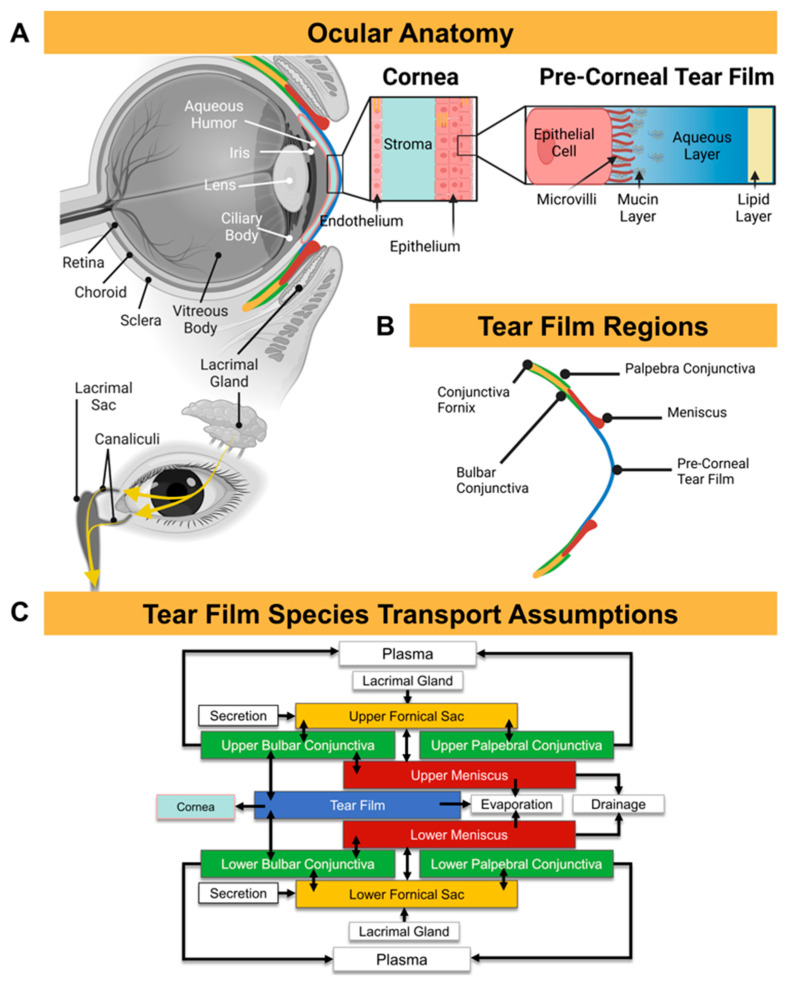
(**A**) Ocular anatomy highlighting the tear film complexity. (**B**) Tear film subregions. (**C**) Tear film species transport assumptions.

**Figure 2 bioengineering-12-00825-f002:**
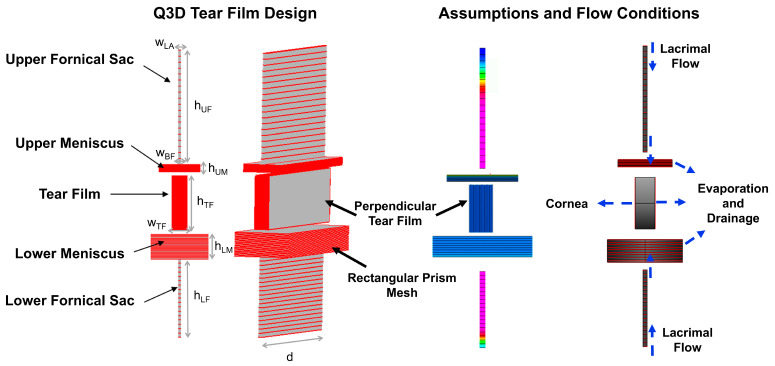
Q3D tear film was constructed using rectangular prisms defined based on anatomical geometric parameters. The fornical sac mesh was aligned with the menisci, while the tear film prisms were oriented perpendicularly. Flow assumptions are indicated by the blue arrows, allowing for simulations of drug permeability and hydrodynamics affected by blinking, drainage, and evaporation.

**Figure 3 bioengineering-12-00825-f003:**
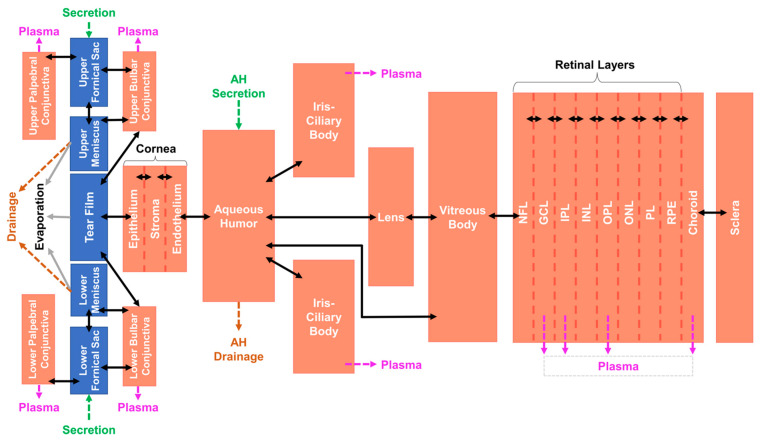
Compartmental model assumptions describing species transport for the whole eye, containing the vitreous body and retinal layers. AH = aqueous humor; NFL, GCL, IPL, INL, OPL, PL, and RPE = layers of the retina. The Q3D portion of the model is represented by blue domains.

**Figure 4 bioengineering-12-00825-f004:**
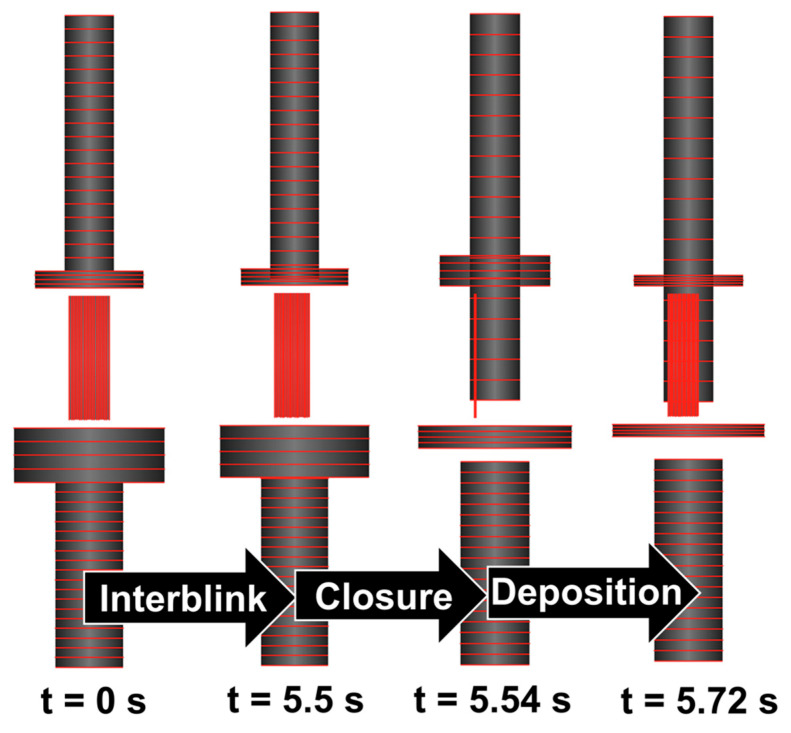
Q3D tear film model blink cycle.

**Figure 5 bioengineering-12-00825-f005:**
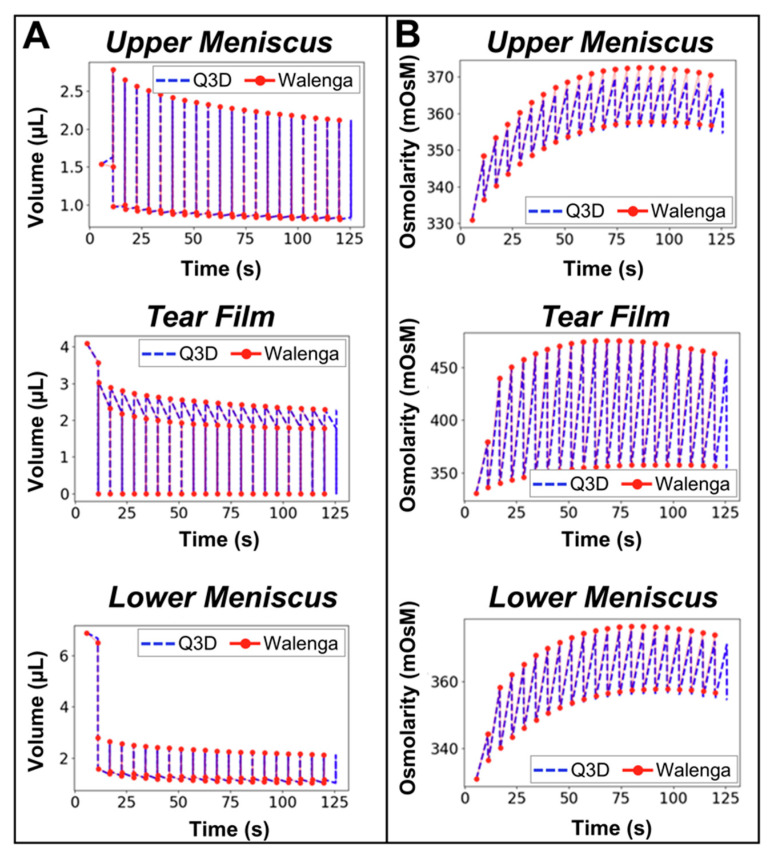
Verification of (**A**) volume and (**B**) osmolarity dynamics over 20 blink cycles comparing Q3D results in this study to re-created results published by Walenga et al. [34].

**Figure 6 bioengineering-12-00825-f006:**
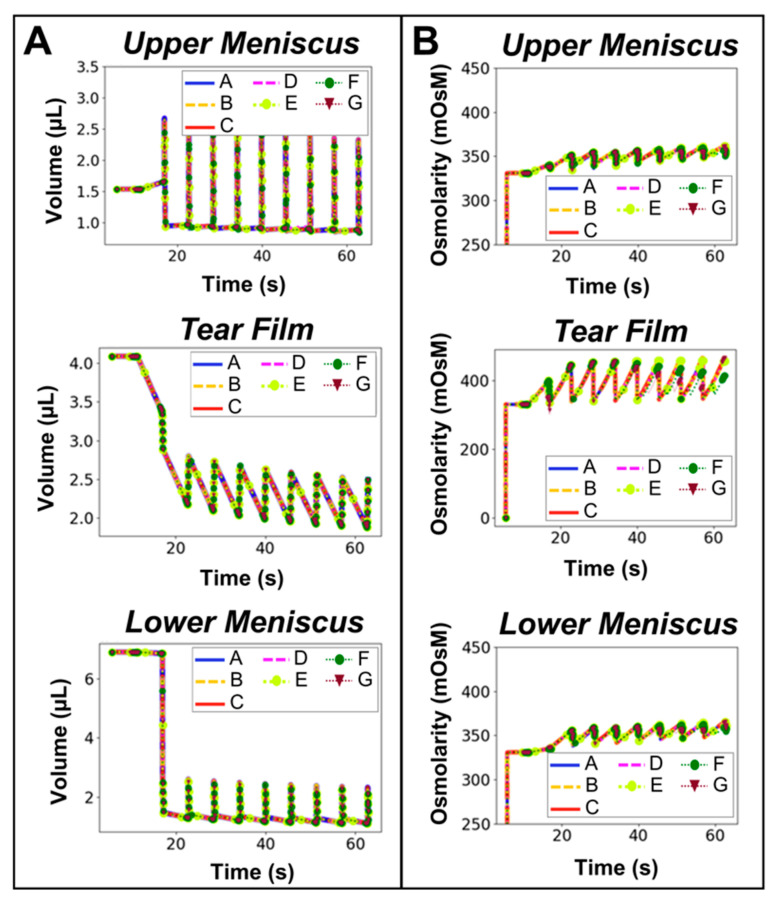
Comparison of Q3D model (**A**) volume and (**B**) osmolarity dynamics over 10 blink cycles for variable and periodic timesteps defined in Table 5.

**Figure 7 bioengineering-12-00825-f007:**
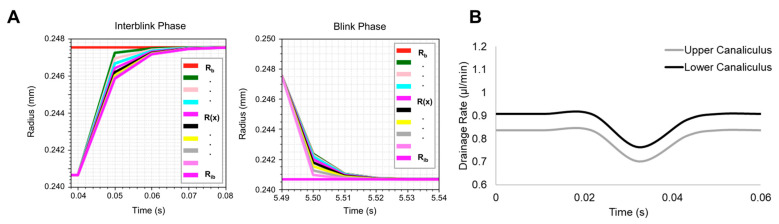
(**A**) Representative upper canaliculi radius profile during blink cycle. The radii reach a steady state during the interblink and blink phases along the length of the canaliculi. This process repeats for all the blink cycles. (**B**) Representative fluctuation in the drainage rate during a single blink ranged from 0.7 to 0.9 µL/min, with a greater drainage rate in the lower canaliculus compared with the upper canaliculus.

**Figure 8 bioengineering-12-00825-f008:**
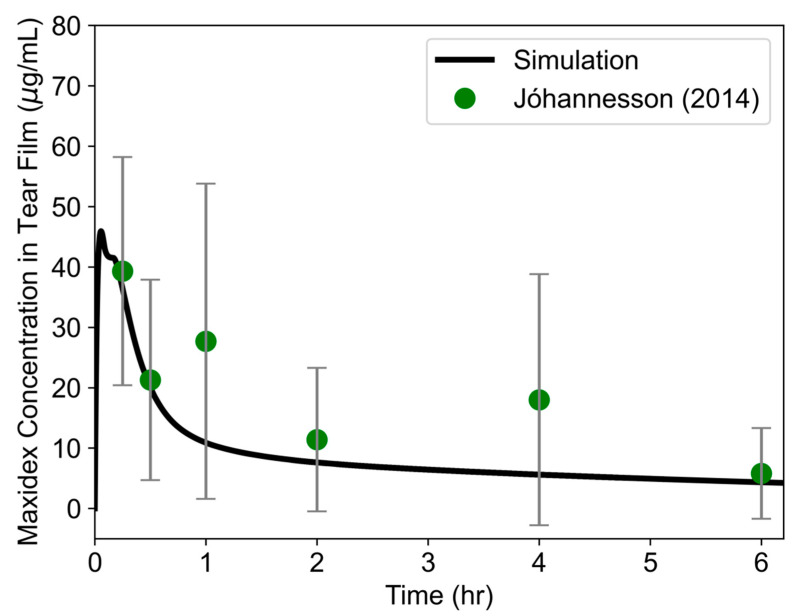
Concentration–time profile of 0.1% dexamethasone (Maxidex^®^ eye drops) in the tear film. Simulated concentrations in the human eye model are compared with experimental data [56].

**Table 1 bioengineering-12-00825-t001:** The Q3D tear film and canaliculi anatomical geometric parameter assumptions based on the literature.

Description	Parameter	Value	Reference
Exposed eye surface area	SA_exp_	220 mm^2^	[32]
Tear film width	w_TF_	18.59 µm	[33,34]
Eyelid length	L_lid_	57 mm	[35]
Initial tear film volume	V_TF0_	4.09 mm^3^	Calculated; SA_exp_ × w_TF_
Tear film depth	d	28.5 mm	Calculated; L_lid_/2
Upper tear film height	h_TF_	7.72 mm	Calculated; SA_exp_/d
Upper fornical sac height	h_UF_	14.1 mm	[36]
Upper meniscus height	h_UM_	0.934 mm	[33,34]
Lower meniscus height	h_LM_	3.02 mm	[33,34]
Lower fornical sac height	h_LF_	10.2 mm	[36]
Upper meniscus surface area	SA_UM_	0.0539 mm^2^	[33,34]
Lower meniscus surface area	SA_LM_	0.242 mm^2^	[33,34]
Lacrimal duct entrance width	w_LA_	2 µm	[37]
Base of fornical sac width	w_BF_	7 µm	[37]
Undeformed canaliculi radius	R_0_	0.25 mm	[38]
Canaliculi length	L_c_	0.012 m	[39]

**Table 2 bioengineering-12-00825-t002:** Drainage parameter assumptions.

Description	Parameter	Value	Reference
Canalicular wall thickness × elastic modulus	bE	2.57 Pa-m	[47]
Eyelid closure time	t_blink_	0.04 s	[43]
End of interblink phase	t_interblink_	5.54 s	[40]
Initial pressure in canaliculi	$p_{0}$	400 Pa	[48]
Pressure in the lacrimal sac	$p_{sac}$	0 Pa (atmospheric)	[46]
Viscosity of the instilled fluid	*μ*	0.0015 Pa-s	[46]

**Table 3 bioengineering-12-00825-t003:** Drainage steady-state assumptions.

Steady-State Assumptions Governing Drainage		
Phase	Blink Phase	Interblink Phase
Limits	0 < t < t_blink_	t_blink_ < t < t_blinkcycle_
Radius	$Rib\left(x,t=0\right)=\frac{R_{0}}{1+\frac{\frac{\sigma }{R_{m}}R_{0}}{bE}}$	$Rb\left(x,t=t_{blink}\right)=\frac{R_{0}}{1+\frac{\left(p_{0}-p_{sac}\right)R_{0}}{bE}}$
Flow rate	q(x=0,t)=0	$q(x=L_{c},t)=0$
Pressure	$p(x=L_{c},t)=0$	$p(x=0,t)=-\frac{\sigma }{R_{m}}$

q = flow rate of tears through the canaliculi; $L_{c}$ = canaliculi length; Rib and Rb = steady-state radii at the end of the blink and interblink phases; $p_{0}$ = initial pressure; $p_{sac}$ = pressure in the lacrimal sac; *σ* = surface tension of the pre-corneal tears; *R_m_* = radius of curvature of the tear meniscus.

**Table 4 bioengineering-12-00825-t004:** Maxidex^®^ (0.1% dexamethasone) suspension inputs.

Description	Parameter	Maxidex^®^
Drug concentration [mg/mL]	C	1
Solubility limit [µg/mL]	Cs	100
Octanol–water partition coefficient	LogP	1.83
Drug mass [mg]	M_drug_	0.035
Drug molecular weight [g/mol]	MW	392.5
Drug density [kg/m^3^]	*ρ*	1300
Distribution coefficient	K_D_	2.5
Particle radius [µm]	*r*	1.5

**Table 5 bioengineering-12-00825-t005:** CPU time for the Q3D blinking human tear film model.

Case Label ^a^	Timestep	Q3D CPU Time	Total No. of Timesteps
A	Uniform dt = 0.001	13 m 47 s	57,200
B	$dt=\left\{\begin{array}{c} 0.001\;\mathrm{if}\;0.00<t<0.01\\ 0.005\;\mathrm{if}\;0.01<t<5.40\\ 0.001\;\mathrm{if}\;5.40<t<5.72 \end{array}\right.$	3 m 53 s	14,072
C	$dt=\left\{\begin{array}{c} 0.001\;\mathrm{if}\;0.00<t<0.01\\ 0.010\;\mathrm{if}\;0.01<t<5.40\\ 0.001\;\mathrm{if}\;5.40<t<5.72 \end{array}\right.$	2 m 13 s	8673
D	$dt=\left\{\begin{array}{c} 0.001\;\mathrm{if}\;0.00<t<0.01\\ 0.050\;\mathrm{if}\;0.01<t<5.40\\ 0.001\;\mathrm{if}\;5.40<t<5.72 \end{array}\right.$	1 m 09 s	4281
E	$dt=\left\{\begin{array}{c} 0.001\;\mathrm{if}\;0.00<t<0.01\\ 0.100\;\mathrm{if}\;0.01<t<5.30\\ 0.001\;\mathrm{if}\;5.30<t<5.72 \end{array}\right.$	1 m 02 s *	3741
F	$dt=\left\{\begin{array}{c} 0.001\;\mathrm{if}\;0.00<t<0.01\\ 0.500\;\mathrm{if}\;0.01<t<4.50\\ 0.001\;\mathrm{if}\;4.50<t<5.72 \end{array}\right.$	2 m 06 s	7800
G	$dt=\left\{\begin{array}{c} 0.001\;\mathrm{if}\;0.00<t<0.01\\ 1.000\;\mathrm{if}\;0.01<t<4.01\\ 0.001\;\mathrm{if}\;4.01<t<5.72 \end{array}\right.$	1 m 58 s	8263

* Best computational performance i.e., lowest CPU time. ^a^ Outputs for each case label can be visualized in Figure 6.

## Data Availability

The original contributions presented in this study are included in the article/Appendix A. Further inquiries can be directed to the corresponding author.

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
