# Peer review of "Quasi-3D Mechanistic Model for Predicting Eye Drop Distribution in the Human Tear Film"

_bioengineering, 2025, doi:10.3390/bioengineering12080825_

Round 1

Reviewer 1 Report

Comments and Suggestions for Authors
  1. The authors should describe how this Q3D model differs from previous Q3D applications in the introduction section. What is the research gap, and how does this method provide a novelty over the other methods?
  2. The authors are advised to provide a summary table that compares Q3D, FEM, and PBPK models. The parameters, like speed, accuracy, resolution, complexity, etc., must be compared.
  3. I feel that the assumption of the constant evaporation rate made by the authors could lead to either over- or under-estimation under dynamic physiological conditions. This should be justified in the discussion section.
  4. It is advised that the authors should discuss how anatomical or physiological variability among patients (e.g., blink rate, tear film volume) might affect model predictions. Have they taken into account such variability in this study? If not, this must be included in the conclusion as a weakness of the study.
  5. In the conclusion section, the authors should explain how the model could be useful for clinicians (e.g., in personalized medicine).
  6. A discussion on the generalizability of the proposed model for children, elderly individuals, or those with ocular surface disorders may be discussed in the conclusion section.

Author Response

Thank you for the opportunity to revise the manuscript titled “Quasi-3D Mechanistic Model for Predicting Eye Drop Distribution in the Human Tear Film” and resubmit to Bioengineering. We understand and greatly appreciate the commitment from the journal as well as the reviewers in order to provide such valuable feedback in a timely manner. The requested revisions have been incorporated into the manuscript and are indicated in red within the document. Detailed responses to the reviewers are below.

Reviewer 1

Comment 1: The authors should describe how this Q3D model differs from previous Q3D applications in the introduction section. What is the research gap, and how does this method provide a novelty over the other methods?

Response:  To our knowledge, the only other Q3D model of the eye was developed by our team and was localized to the cornea to simulate drug transport across the in vitro cornea, which was described in the Introduction. We have adjusted the following sentences to underline the advantage of building a mechanistic model for the tear film compared to other methods:

“Using a similar approach, a Q3D model of the cornea was developed to simulate drug transport across the in vitro cornea where the model was shown to provide both efficient and accurate predictions of lipophilic (RhB) and hydrophilic (fluorescein) tracer distribution throughout the cornea [21]. However, the accuracy of predictions of corneal distribution are directly dependent on the accuracy of tear film distribution. This can be especially important when investigating the effects of ocular conditions on drug distribution. Building upon this foundational work, the purpose of this study was to develop a fast-running, multi-physics Q3D-based model of the tear film to simulate drug administration, drainage, tear evaporation, and spatial drug distribution, accounting for blinking effects.”

Comment 2: The authors are advised to provide a summary table that compares Q3D, FEM, and PBPK models. The parameters, like speed, accuracy, resolution, complexity, etc., must be compared.

Response: We appreciate the reviewer's valuable feedback regarding the need for a comparative analysis of modeling approaches. We acknowledge that such a comparison would enhance the manuscript's clarity for readers.

To clarify our terminology, PBPK (physiologically-based pharmacokinetic) models represent compartmental approaches that utilize ordinary differential equations (ODEs) to characterize drug transport between different physiological compartments (such as organs and tissues). Conversely, Q3D (quasi-three-dimensional) and FEM (finite element method) approaches employ partial differential equations (PDEs) to capture both spatial and temporal phenomena, making them particularly valuable for applications demanding spatial detail (such as analyzing drug penetration through skin layers or tissue structures).

Following the reviewer's recommendation, we have incorporated a comprehensive qualitative comparison table that outlines the fundamental differences between Q3D, FEM, and PBPK (ODE-based) modeling strategies. This comparison encompasses critical factors including computational efficiency, spatial accuracy, implementation difficulty, computational demands, visualization features, and suitable applications. The table effectively demonstrates the balance between model sophistication and computational performance across these three methodologies.

Parameter

Q3D

PBPK (ODE)

FEM

Mathematical Foundation

Uses partial differential equations (PDE)

Uses ordinary differential equations (ODE)

Uses partial differential equations (PDE)

Model Development Complexity and Time

Moderate: CoBi platform simplifies setup, minimal manual coding of equations

Low to High: simple for basic models, complex when discretization needed (100s of equations/loops)

High: complex mesh setup and implementation

Computational Speed

Moderate: faster than FEM, slightly slower than ODEs

Fast: fastest among the three approaches

Slow: most computationally intensive

Computational Resources

Moderate to low resource requirements

Low resource requirements

High resource requirements

Mesh Setup

Mesh setup is simple and user friendly

Not applicable (no spatial discretization)

Manual mesh setup is difficult and time-consuming

Accuracy

Good spatial resolution with quasi-3D approximations

Limited spatial resolution, suitable for lumped parameter models

Highest accuracy with full 3D spatial resolution

Visualization Capabilities

Strong visualization capabilities

Limited visualization (typically 1D plots)

Excellent 3D visualization capabilities

Multiphysics Coupling

Easier coupling compared to manual approaches, using CoBi

Manual coupling required - more complex to implement

Easier coupling within FEM framework

Scalability

Good scalability and works well for extended domains and layered tissues

Excellent for simple models, limited for spatially discretized problems

Limited scalability due to computational demands

User Expertise Required

Moderate: Requires some platform familiarity.

Low to High: Depends on model complexity

High: Requires FEM expertise and mesh generation skills

Best Use Cases

Problems with skewed aspect ratios (similar to 1D geometry), moderate complexity requiring spatial detail with computational efficiency

Simple lumped parameter models, rapid prototyping

Complex 3D geometries, high-accuracy applications requiring full 3D spatial resolution where FEM complexity is justified

The authors have decided to include this table with qualitative comparisons to the Supplemental Information so as not to detract from the main focus of the manuscript. The following sentence was added to the Introduction:

“A detailed table comparing PBPK, FEM, and Q3D modeling strategies can be found in the Supplemental Information (Table S1).”

Comment 3: I feel that the assumption of the constant evaporation rate made by the authors could lead to either over- or under-estimation under dynamic physiological conditions. This should be justified in the discussion section.

Response: This is a good point. The sentence “This assumption could lead to over- or under-estimation under dynamic physiological conditions.” has been added to the discussion. We then provide citations for how this limitation could be improved in future model iterations using physics-based models.

Comment 4: It is advised that the authors should discuss how anatomical or physiological variability among patients (e.g., blink rate, tear film volume) might affect model predictions. Have they taken into account such variability in this study? If not, this must be included in the conclusion as a weakness of the study.

Response: Thank you for this suggestion. The developed model can be tailored for subject-specific anatomy and physiology, which is an advantage. However, validation has only been performed for healthy adult populations, which limits the functionality at present. The following sentences have been added to the discussion to address these drawbacks:

“The complex geometry of the human eye and minimal datasets quantifying its complex anatomical features, particularly in the tear film, led to an approach centered around conservation of volume within rectangular prism substructures. However, anatomical or physiological variability among patient populations (e.g. blink rate, tear volume) inherently limits the generalizability of this model beyond that of a healthy adult. Q3D tear film model performance is directly tied to the accuracy of the geometric and physiological parameter assumptions, which we expect will continue to converge as technological innovations improve quality of experimental assessments over time.”

Comment 5: In the conclusion section, the authors should explain how the model could be useful for clinicians (e.g., in personalized medicine).

Response: Thank you for this suggestion. We have added the following sentence to the conclusion:

“Q3D tear film modeling could be a powerful tool for clinicians to recommend optimal dosing based on patient-specific parameters (i.e. geometry, drainage, blink rates) as a step towards improved personalized medicine.”

Comment 6: A discussion on the generalizability of the proposed model for children, elderly individuals, or those with ocular surface disorders may be discussed in the conclusion section.

Response: This is a great point, which was very similar to Reviewer 2, Comment 9. We have added the following sentences in the discussion section.

“While this work demonstrates the novelty and feasibility of Q3D implementation within the tear film, further validation should be conducted to ensure reliable prediction of drug distribution throughout various regions of the eye. This will require a large amount of experimental or literature data, which is mostly available for animals rather than humans. Additionally, the developed Q3D model was built based on experimentally derived parameters from healthy adults. The lack of available datasets for children and elderly individuals as well as those with ocular surface disorders limits applications of this model. To establish a well-defined verification and validation protocol and expand the clinical applications of these models, the authors are working on compiling all available data into a single open-source public repository.”

Reviewer 2 Report

Comments and Suggestions for Authors

This manuscript presents a model with potential to advance ocular drug delivery modeling. However, additional clarity, are needed to fully realize its impact. Addressing the issues above will significantly strengthen the manuscript and improve its utility for both academic and translational audiences.

  1. Title and abstract look good.
  2. Clarify and expand the input parameters (e.g., eye surface area, tear volume, blinking frequency) and their sources (literature vs. experimental).
  3. Briefly describe the numerical methods used to solve the ODE system (e.g., solver type, time steps).
  4. Explain the rationale for using a quasi-3D approach—how does it differ from full 3D and 2D models in complexity and accuracy?
  1. Please indicate how well these metrics compare to other published ocular PK models.
  2. Describe the experimental data source and the sampling strategy used to validate the model.
  3. Please justify the choice of 0.1% dexamethasone—is this representative of typical topical formulations?
  4. Discuss whether blinking is implemented as a periodic mechanical event or a dynamic fluid shift, and how this affects drug distribution.
  5. The manuscript does not sufficiently address the assumptions and limitations of the model. Discuss how these assumptions may affect model generalizability.
  6. State the hardware specifications used for testing computational performance.
  7. Suggest use cases such as predicting systemic exposure from ocular delivery, especially for potent corticosteroids like dexamethasone.
  8. Elaborate on how ocular absorption data from this model could be incorporated into systemic models (e.g., via tear-to-blood transport compartments).
  9. Comment on how this tool could support in silico bioequivalence studies for generic ophthalmic drugs.
  10. Discuss the potential for this model to assist in topical formulation optimization or regulatory decision-making.
  11. The following studies are suggested to evaluate and add in the literature review of manuscript: https://doi.org/1021/acsmaterialslett.4c02327, https://doi.org/10.7150/thno.104752, https://doi.org/10.1007/s10637-024-01464-w

Author Response

Thank you for the opportunity to revise the manuscript titled “Quasi-3D Mechanistic Model for Predicting Eye Drop Distribution in the Human Tear Film” and resubmit to Bioengineering. We understand and greatly appreciate the commitment from the journal as well as the reviewers in order to provide such valuable feedback in a timely manner. The requested revisions have been incorporated into the manuscript and are indicated in red within the document. Detailed responses to the reviewers are below.

Reviewer 2

This manuscript presents a model with potential to advance ocular drug delivery modeling. However, additional clarity, are needed to fully realize its impact. Addressing the issues above will significantly strengthen the manuscript and improve its utility for both academic and translational audiences.

Comment 1: Title and abstract look good.

Response: Thank you.

Comment 2: Clarify and expand the input parameters (e.g., eye surface area, tear volume, blinking frequency) and their sources (literature vs. experimental).

Response: Most input parameters were based on experimentally determined values found in published literature. References for these inputs and assumptions are provided in the tables as well as the text. If a parameter was estimated, such as the calculated tear film depth and height, equations are provided. See Table 1 and Table S4 for details.

Comment 3: Briefly describe the numerical methods used to solve the ODE system (e.g., solver type, time steps).

Response: Standard solvers, such as Euler method, Runge-Kutta, and Stiff solvers are implemented into CoBi software to solve systems of ODEs. The platform provides multiple solver options including explicit methods (Euler, Runge-Kutta) for non-stiff systems and specialized stiff solvers for systems with widely varying time scales. This information has been added to the methods.

Due to the linking of the ODE model (posterior eye) with the Q3D model, the ODE time steps were limited by the smallest blinking phase/eyelid closure time (0.04 s). Therefore, to capture changes during this period, the time step was set to 0.01 s.

Comment 4: Explain the rationale for using a quasi-3D approach—how does it differ from full 3D and 2D models in complexity and accuracy?

Response: The following rationale was added to the discussion:

“The quasi-3D (Q3D) approach was selected to balance computational efficiency with spatial accuracy for ocular drug delivery applications characterized by highly skewed aspect ratios, where one dimension significantly exceeds the others (e.g., thin tear film layers, corneal epithelium). Q3D models offer intermediate complexity between 2D and full 3D approaches—capturing essential three-dimensional effects through dimensional reduction techniques while avoiding the computational burden of full 3D finite element discretization. While full 3D models provide the highest spatial fidelity, Q3D achieves comparable accuracy for geometries with skewed aspect ratios at significantly reduced computational cost, effectively capturing dominant transport phenomena (diffusion across tissue layers, convective flow in tear film) without significant loss of physical accuracy.”

Altogether, this approach enables realistic modeling of complex physiological processes while maintaining computational efficiency suitable for parameter studies and clinical applications.

Comment 5: Please indicate how well these metrics compare to other published ocular PK models.

Response: The model fit was compared to other PK models where it was stated that “Simulation of Maxidex® suspension fell within the range of experimental tear film concentrations at each time point where R2 was 0.76. This measure accounts for variability in experimental data where 0.6-0.9 is considered a good fit in PBPK models [64].”

[52] M. Le Merdy, J. Fan, M.B. Bolger, V. Lukacova, J. Spires, E. Tsakalozou, V. Patel, L. Xu, S. Stewart, A. Chockalingam, Application of mechanistic ocular absorption modeling and simulation to understand the impact of formulation properties on ophthalmic bioavailability in rabbits: a case study using dexamethasone suspension, AAPS J. 21 (2019) 1–11.

Comment 6: Describe the experimental data source and the sampling strategy used to validate the model.

Response: We have expanded our methods to include the following in Section 2.7:

“Validation data were published in Jóhannesson et al. where 0.1% dexamethasone suspension (Maxidex ®) concentration was quantified in the tear film of six healthy volunteers at 0.25, 0.5, 1, 2, 4, 6, and 24 hours following topical administration. A micropipette capillary tube was placed in the tear meniscus inside the lower eye lid to collect approximately 5 ul of tear fluid.”

Comment 7: Please justify the choice of 0.1% dexamethasone—is this representative of typical topical formulations?

Response: Yes, suspensions are commonly used formulations. The choice to model 0.1% dexamethasone formulation was based on the available published datasets, which was stated in the discussion. Datasets for this formulation (Maxidex) are available in vitro, in vivo, and across species, which makes it great for validation. This was reiterated in the discussion:

“Dexamethasone formulations have been on the market for decades, so sufficient data in humans allowed for effective validation and assessment of drug-specific model performance.”

Comment 8: Discuss whether blinking is implemented as a periodic mechanical event or a dynamic fluid shift, and how this affects drug distribution.

Response: One of the advantages of the CoBi platform is that it allows for coupling of these mechanisms. Blinking was implemented as a periodic mechanical event, but estimations of the dynamic fluid shifts were calculated within each period.

Comment 9: The manuscript does not sufficiently address the assumptions and limitations of the model. Discuss how these assumptions may affect model generalizability.

Response: Thank you for allowing us the opportunity to expand on these assumptions and limitations, which was very similar to Reviewer 1, Comment 6. We feel that we have now expanded on many of the main limitations, especially the generalizability of this model. Given that this model was built based on data and experimentally-derived parameters from healthy human adult populations, accuracy of model predictions is expected to be lower in pediatric and elderly populations as well as those with ocular surface disorders. The following sentences were included in the discussion to underline this limitation:

“Note that despite these error values, this model for drug distribution throughout the tear film was only evaluated based on a dataset from six healthy adults and more robust, population-based assessments of model performance are necessary prior to implementation of this tool in clinical settings.”

“Additionally, the developed Q3D model was built based on experimentally derived parameters from healthy adults. The lack of available datasets for children and elderly individuals as well as those with ocular surface disorders limits applications of this model. To establish a well-defined verification and validation protocol and expand the clinical applications of these models, the authors are working on compiling all available data into a single open-source public repository.”

Comment 10: State the hardware specifications used for testing computational performance.

Response: The hardware specifications were expanded and are listed in Section 3.2 as follows:

“All simulations were performed on personal computing laptops (Dell) with 64-bit Windows operating systems. CPU was computed on devices equipped with 16GB of RAM. Please note that CoBi modeling platform is OS-agnostic and not restricted to Windows operating systems or any specific hardware manufacturer.” 

Comment 11: Suggest use cases such as predicting systemic exposure from ocular delivery, especially for potent corticosteroids like dexamethasone.

Comment 12: Elaborate on how ocular absorption data from this model could be incorporated into systemic models (e.g., via tear-to-blood transport compartments).

Comment 14: Discuss the potential for this model to assist in topical formulation optimization or regulatory decision-making.

Response: We have incorporated suggestions from Comments 11, 12, and 14 into the following paragraph within the discussion:

Assessment and validation of drug clearance throughout the posterior eye and whole-body systems could enable use of the developed model for prediction of systemic safety in future iterations. Areas for future development include transport of tear fluid into the blood stream, which can occur through the lacrimal glands, the blood-retinal barrier, or interaction between blood vessels and outer ocular regions [66]. Expansion of model complexity to include these transport mechanisms can enable optimization of topical formulations to reduce the likelihood of adverse events from ocular delivery. This capability is advantageous, especially for potent corticosteroids like dexamethasone, which can contribute to a wide range of systemic complications [67].”

Comment 13: Comment on how this tool could support in silico bioequivalence studies for generic ophthalmic drugs.

Response: The following sentence was added to the discussion:

“With access to larger, open-source datasets, ocular modeling frameworks may be used to perform virtual bioequivalence studies to support the development of generic ophthalmic drug products.”

Comment 15: The following studies are suggested to evaluate and add in the literature review of manuscript: https://doi.org/1021/acsmaterialslett.4c02327 , https://doi.org/10.7150/thno.104752 , https://doi.org/10.1007/s10637-024-01464-w

Response: While these studies on 1) a therapeutic for dry eye, 2) analysis of neuroinflammatory mechanisms at the blood-retinal barrier, and 3) assessment of pembrolizumab-induced uveitis clinical characteristics significantly contribute to the field, we felt that these studies were outside the scope of the current manuscript.

Reviewer 3 Report

Comments and Suggestions for Authors

The authors presented the development of a quasi-3D mechanistic model that represents tear film. The manuscript is well-written and here are some suggestions that could improve it:

  1. please correct the abstract, because not a human eye model was created, but a tear model was (lines 17 and 18);
  2. I think "dry eye" is not an adequate keyword for this manuscript;
  3. in the introduction please add some details (with appropriate references) regarding ODE, PDE and FEM;
  4. Line 205: I suppose it should be "rheological" instead of "theological";
  5. Lines 286 and 287: at least laptop manufacturer and operating system should be added;
  6. Table 5: explain the case labels with more details;

Author Response

Thank you for the opportunity to revise the manuscript titled “Quasi-3D Mechanistic Model for Predicting Eye Drop Distribution in the Human Tear Film” and resubmit to Bioengineering. We understand and greatly appreciate the commitment from the journal as well as the reviewers in order to provide such valuable feedback in a timely manner. The requested revisions have been incorporated into the manuscript and are indicated in red within the document. Detailed responses to the reviewers are below.

Reviewer 3

The authors presented the development of a quasi-3D mechanistic model that represents tear film. The manuscript is well-written and here are some suggestions that could improve it:

Comment 1: please correct the abstract, because not a human eye model was created, but a tear model was (lines 17 and 18);

Response: This has been corrected from “human eye” to “human tear film”.

Comment 2: I think "dry eye" is not an adequate keyword for this manuscript;

Response: The keyword “dry eye” has been removed and additional keywords such as “fast-running”, “blinking”, and “drainage” have been added. Thank you for this suggestion.

Comment 3: in the introduction please add some details (with appropriate references) regarding ODE, PDE and FEM;

Response: Ocular models using PBPK, FEM, and Q3D methods were cited in the Introduction. However, we now include additional information comparing critical factor of each method including computational efficiency, spatial accuracy, implementation difficulty, computational demands, visualization features, and suitable applications. These are listed in the response to Reviewer 1, Comment 2 as well as the supplemental information within the manuscript.

Comment 4: Line 205: I suppose it should be "rheological" instead of "theological";

Response: Yes, thank you for finding this typo. It is now corrected.

Comment 5: Lines 286 and 287: at least laptop manufacturer and operating system should be added;

Response: The laptop manufacturer and operating system have been added and the sentences now read:

“All simulations were performed on personal computing laptops (Dell) with 64-bit Windows operating systems. CPU was computed on devices equipped with 16GB of RAM. Please note that CoBi modeling platform is OS-agnostic and not restricted to Windows operating systems or any specific hardware manufacturer.”

Please note that while our simulations were conducted on a Dell laptop running Windows, CoBi is platform-independent and not restricted to Windows operating systems or any specific hardware manufacturer. The reported CPU times, obtained using a 64-bit Windows system with 16GB of RAM, provide a representative benchmark that enable readers to interpret computational performance across different computing environments.

Comment 6: Table 5: explain the case labels with more details;

Response: The “Case Label” provides a reference for the outputs shown in Figure 6. This is now indicated in Table 5.

Reviewer 4 Report

Comments and Suggestions for Authors

In this manuscript, Garimella et al. discuss a new Quasi-3D (Q3D) model that was developed to predict the distribution of eye drops through the tear film in the eye. The authors provide good reasoning thoughtfully developed methods used for establishing this model and also compared predicted data from the model against real data available for a Dexamethasone-based commercially available eye drop. Such models are useful for the field as dry eye conditions become more common, and start affecting the quality of life of individuals. This model can help predict the potential for success for eye drops being formulated.

This is a well-written manuscript with well thought-out methods. Minor comments:

  1. For each Table, please add column heading ‘Parameter symbol’ for the column where symbols are listed.
  2. Instead of ‘Citation’, label the column header as ’Reference’.

Author Response

Thank you for the opportunity to revise the manuscript titled “Quasi-3D Mechanistic Model for Predicting Eye Drop Distribution in the Human Tear Film” and resubmit to Bioengineering. We understand and greatly appreciate the commitment from the journal as well as the reviewers in order to provide such valuable feedback in a timely manner. The requested revisions have been incorporated into the manuscript and are indicated in red within the document. Detailed responses to the reviewers are below.

Reviewer 4

In this manuscript, Garimella et al. discuss a new Quasi-3D (Q3D) model that was developed to predict the distribution of eye drops through the tear film in the eye. The authors provide good reasoning thoughtfully developed methods used for establishing this model and also compared predicted data from the model against real data available for a Dexamethasone-based commercially available eye drop. Such models are useful for the field as dry eye conditions become more common, and start affecting the quality of life of individuals. This model can help predict the potential for success for eye drops being formulated.

Comment 1: This is a well-written manuscript with well thought-out methods.

Response: Thank you.

Comment 2: For each Table, please add column heading ‘Parameter symbol’ for the column where symbols are listed.

Response: We agree that there needs to be a separate heading between the “Description” and “Parameter”. We now include these terms to match the supplemental table headings.

Comment 3: Instead of ‘Citation’, label the column header as ’Reference’.

Response: Tables 1 and 2 have been adjusted.

Round 2

Reviewer 1 Report

Comments and Suggestions for Authors

Accept in present form

Reviewer 2 Report

Comments and Suggestions for Authors

All queries resolved. The revision can be accepted 

Reviewer 3 Report

Comments and Suggestions for Authors

The authors corrected manuscript in accordance with my comments and I think It can be accepted for publication